# 🪄 THE PENSIEVE PARADIGM: STATEFUL LANGUAGE MODELS MASTERING THEIR OWN CONTEXT

**Xiaoyuan Liu**[1,2*] **Tian Liang**[2] **Dongyang Ma**[2] **Deyu Zhou**[3] **Haitao Mi**[2]
**Pinjia He**[1†] **Yan Wang**[2]
[1]School of Data Science, The Chinese University of Hong Kong, Shenzhen
[2]Tencent AI Lab   [3]Hong Kong University of Science and Technology, Guangzhou
`xyliu.cs@gmail.com, yanwang.branden@gmail.com`
`hepinjia@cuhk.edu.cn`

## ABSTRACT

In the world of Harry Potter, when Dumbledore's mind is overburdened, he extracts memories into a Pensieve to be revisited later. In the world of AI, while we possess the Pensieve—mature databases and retrieval systems, our models inexplicably lack the "wand" to operate it. They remain like a Dumbledore without agency, passively accepting a manually engineered context as their entire memory. This work finally places the wand in the model's hand. We introduce StateLM, a new class of foundation models endowed with an internal reasoning loop to manage their own state. We equip our model with a suite of memory tools, such as context pruning, document indexing, and note-taking, and train it to actively manage these tools. By learning to dynamically engineering its own context, our model breaks free from the architectural prison of a fixed window. Experiments across various model sizes demonstrate StateLM's effectiveness across diverse scenarios. On long-document QA tasks, StateLMs consistently outperform standard LLMs across all model scales; on the chat memory task, they achieve absolute accuracy improvements of 10% to 20% over standard LLMs. On the deep research task BrowseComp-Plus, the performance gap becomes even more pronounced: StateLM achieves up to 52% accuracy, whereas standard LLM counterparts struggle around 5%. Ultimately, our approach shifts LLMs from passive predictors to state-aware agents where reasoning becomes a stateful and manageable process.

## 1 INTRODUCTION

A fundamental limitation of current LLMs is their stateless, autoregressive nature. At their core, they are passive predictors, architecturally designed to perform sequence completion within an externally-provided context. This forces them to operate like a mind with no long-term memory of its own, unable to actively manage or alter their own reasoning process. This core limitation has pushed the area towards brittle, external workflows of "Context Engineering", where complex reasoning is offloaded to human-written scripts that orchestrate a series of isolated LLM calls. Whether through brute-force

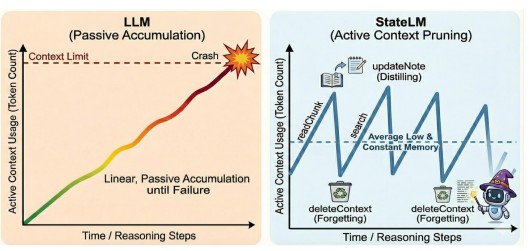

Figure 1: StateLM (right) maintains a "sawtooth" context-use profile, rather than monotonic accumulation (left).

context window expansion or scripted workflows, the model itself remains a passive component without the capability to strategize or manage its own memory.

---

* Work was done when interned at Tencent AI Lab.
† Correspondence to: Pinjia He <hepinjia@cuhk.edu.cn>.

This dilemma evokes a powerful analogy from the world of *Harry Potter*. With a touch of his wand, Dumbledore extracts, stores, and organizes his memories in a Pensieve. He possesses complete agency: he decides what to offload, what to revisit, and even how to share his thoughts with others like Harry. This vision of active memory management stands in stark contrast to our current reality in AI. A profound paradox exists: while we have built the Pensieve, the vast databases and retrieval systems, we have never given the model the wand. Instead, we humans, act as the wizards. We use our own "spells" (e.g., context engineering) to pull out what we think is relevant information, and then force-feed it into the model's context. The model is not the wizard; it is merely a passive observer of the magic we perform on its behalf.

The central question of our work is therefore simple: What happens when we finally place the wand in the model's hand? We answer this by introducing **Stateful Language Models** (referred to as **StateLMs**), a new class of foundation models endowed with a learned capability to self-engineer their context. As illustrated in Figure 1, unlike traditional models that accumulate tokens linearly until hitting the limit, StateLM maintains an efficient, "sawtooth" context length. The cornerstone of this mechanism is the *deleteContext* tool, which empowers model the capability to self-forget redundant tokens from its context. By coordinating this with auxiliary "spells" for data ingestion (*readChunk*) and insight distillation (*updateNote*), the model establishes a dynamic reasoning loop: it reads a segment, record key information into persistent notes, and immediately remove the raw text. With this learned **"self-context engineering"** capability, StateLM consistently maintains a compact, noise-free state that allows for sustainable, accurate reasoning regardless of the total context length.

We evaluate StateLM through a set of complementary experiments. On long-document QA benchmarks, StateLM consistently outperforms instruct baselines while using only 1/4 of their active context, with average accuracy gains ranging from 5% to 12% across model scales. StateLM also surpasses prior agentic methods under identical context budget. Beyond document QA, StateLM generalizes to more diverse, complex settings such as long-term chat memory and deep research tasks without task-specific tuning. On the chat memory task, they achieve absolute accuracy improvements of 10% to 20% over standard LLMs. On the deep research task BrowseComp-Plus, the performance gap becomes even more pronounced: The StateLM series achieves up to 52% accuracy, whereas the best vanilla LLM counterparts remain around 5%, resulting in an average improvement of over 40%. Further analysis shows that the improvements are consistent across diverse problem aspects, and that the learned tool-use patterns adapt accordingly to different tasks. Together, these results establish StateLM as a pioneering, general approach for diverse scenarios, shifting LLMs from passive predictors to state-aware systems.

Our contributions are threefold:

- **The Pensieve Paradigm**: We introduce a novel framework that shifts context engineering from external, human-written scripts to the model's own agency, enabling LLMs to actively manage their own context.
- **StateLM**: We develop a new class of foundation models endowed with learned *self-context engineering* capabilities, allowing the model to dynamically manage its reasoning state through a general-purpose toolkit of memory operations.
- **Cross-Domain Generality**: We demonstrate significant performance gains across long-document QA, multi-turn dialogue, and deep research. To our knowledge, this is the first study to simultaneously address the memory challenges of these three domains within a single, unified model.

## 2 RELATED WORK: HUMAN AS THE WIZARD

The core challenge of statelessness has not gone unnoticed. The dominant response, however, has been to cast the human developer as the wizard, meticulously orchestrating the model's limited memory. This paradigm of external control is defined by Andrej Karpathy as "context engineering":

> "...the delicate art and science of filling the context window with just the right information for the next step..." (Karpathy, 2025)

This observation frames the entire landscape of current research. The vast majority of work focuses on building a better context-workflow while the model remains a passive component, waiting for the human wizard to manage its memory.

**RAG**   The most prevalent form of context engineering is Retrieval-Augmented Generation (RAG) (Lewis et al., 2020; Gao et al., 2024). In the standard RAG paradigm, a pipeline is designed around a stateless LLM. The process involves taking a user query, using a dense retriever to search a vector database (the Pensieve) for relevant text chunks, and "stuffing" this retrieved context into the model's prompt. The model has no agency; it passively accepts the context it is given. Recent advancements like ALR$^2$ (Li et al., 2024) and Search-O1 (Li et al., 2025a) make this external workflow more sophisticated, but they still operate within a workflow predefined by the human developer.

**Agentic Memory**   A second line of work moves toward more structured memory systems for LLM agents. Frameworks like MemGPT (Packer et al., 2024) and MemOS (Li et al., 2025b) introduce operating-system-like memory hierarchies to page information in and out of the context window, whereas Memory-R1 (Yan et al., 2025), and MemAgent (Yu et al., 2025) apply the memory update actions at fixed, human-specified points. Most recently, research has shifted toward making these context management behaviors learnable through Reinforcement Learning (RL). *Context-Folding* (Sun et al., 2025a) introduces a framework that empowers agents to manage their context by procedurally branching into sub-trajectories and "folding" them upon completion. This allows the model to collapse intermediate steps while retaining a concise summary. Similarly, *ReSum* (Wu et al., 2025) proposes a paradigm for indefinite exploration by training agents to perform periodic context summarization. By converting interaction histories into compact states, *ReSum* enables models to bypass context constraints.

However, even these advanced methods remain bound by the "Human as the Wizard" paradigm. In *Context-Folding* (Sun et al., 2025b), the model is trained to adapt to a specific, human-designed branching-and-folding routine. In *ReSum* (Wu et al., 2025), the model learns to reason within a fixed summarization schedule. In both cases, the fundamental logic of context management is hard-coded into the training objective or the tool definition.

**This Study: The Model as its Own Context Engineer.**   While existing works train models to **adapt** to human-predefined context engineering routines, StateLM trains the model to **become** its own context engineer. Instead of teaching the model to merely execute a rigid routine like folding or summarizing, we place the wand in its hand. StateLM is equipped with a general-purpose toolkit of memory operations—such as `deleteContext`, `readChunk`, and `updateNote`—and is trained to use them strategically. The model is no longer a passive observer of a predefined workflow; it learns to architect its own reasoning loop, dynamically deciding which information is vital, which is noise, and how to structure its own reasoning state.

## 3   METHODOLOGY

This section introduces our proposed method, StateLM. We first formalize the problem setup (Section 3.1), then describe the our proposed StateLM framework and the Pensieve paradigm (Section 3.2). Finally, we outline the training procedures used to develop the StateLM (Section 3.3).

### 3.1   PROBLEM SETUP

We consider a generic tool-augmented agentic reasoning process for language models. Given a user query $q$, the agent interacts with an environment over a sequence of rounds $t = 1, \ldots, T$. Initially, the state is $s_0 = [q]$. At each subsequent round $t$, the agent conditions on an interaction state

$$s_t = \big[q, a_1, o_1, \ldots, a_{t-1}, o_{t-1}\big],$$

where each $(a_i, o_i)$ denotes a past assistant action and its corresponding environment observation.

Although the interaction is defined over the abstract state $s_t$, the language model operates on a textual prompt obtained by applying a chat templating function $\mathcal{C}$, which serializes the interaction into a token sequence representing the model's reasoning state and injects additional text such as system prompt and tool descriptions $x_t = \mathcal{C}(s_t)$. For clarity, we omit this step in the subsequent discussion.

At round $t$, the agent samples an action $a_t \sim \pi_\theta(\cdot \mid s_t)$. Each action consists of a natural-language reasoning segment ("thought") followed by a tool invocation ("act"). The environment then returns an observation $o_t = \mathcal{E}(s_t, a_t)$, where $\mathcal{E}$ denotes the environment dynamics.

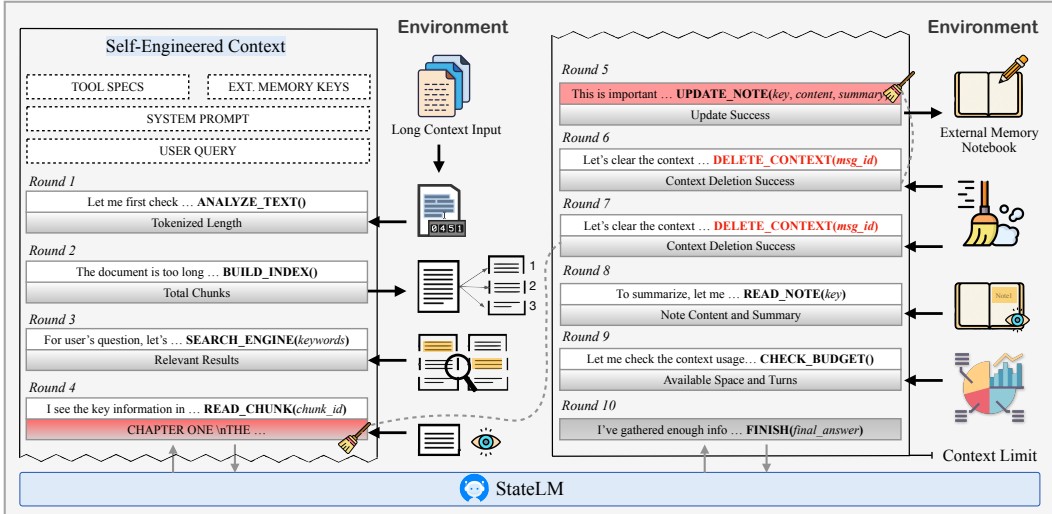

Figure 2: **The self-context engineering workflow of StateLM.** Given a query over a long context, StateLM engages in a multi-round, stateful reasoning loop that analyzes the input, builds an index, and iteratively searches, reads, takes notes, and prunes its working context. Messages highlighted in red are replaced with stubs after the deletion operation. The loop terminates once StateLM determines it has gathered sufficient information for the final answer.

A defining characteristic of this formulation is the **monotonic growth** of the interaction state:

$$s_{t+1} = s_t \,\|\, (a_t, o_t),$$

where $\|$ denotes ordered concatenation. All prior actions and observations are permanently retained and exposed to the model in future rounds. While effective for short-horizon reasoning, this monotonic accumulation becomes a critical limitation in long-horizon settings. Earlier reasoning steps and raw tool outputs persist in the prompt and continuously consume the model's fixed context budget, eventually leading to context exhaustion and performance degradation.

### 3.2 OUR METHOD: STATELM REASONING WITH PENSIEVE

StateLM extends the generic agentic reasoning loop by allowing the model to explicitly modify its own interaction history. Rather than treating the interaction state as an append-only record, we endow the policy with the ability to actively control which past elements remain visible. This transforms the interaction state from a passive log into **a mutable, stateful object** that can be manipulated over time.

$$s_{t+1} = s_t \,\|\, (a_t, o_t) \;\; \rightarrow \;\; s_{t+1} = \mathcal{F}(s_t, a_t, o_t),$$

where $\mathcal{F}$ is a state-evolution function that can append new interactions or modify historical elements based on context-management actions.

Table 1: The StateLM "Spellbook": a general-purpose toolkit for stateful context management.

| Tool Name | Description |
|---|---|
| *Context Perception (Understanding the environment)* | |
| analyzeText | Returns the input length. |
| checkBudget | Reports remaining interaction budget. |
| *Information Acquisition (Accessing raw input)* | |
| buildIndex | Builds a searchable index. |
| searchEngine | Searches for relevant segments. |
| readChunk | Loads a selected text chunk. |
| *Memory Management (Distilling signal and pruning noise)* | |
| note/updateNote | Records or updates key knowledge. |
| readNote | Retrieves stored notes into context. |
| deleteContext | Removes messages from context. |
| *Termination* | |
| finish | Ends reasoning and outputs the answer. |

The core mechanism enabling this transformation for the agentic loop is the *Pensieve paradigm*, which we define as a persistent external memory coupled with explicit context-pruning operations. Conceptually, the Pensieve comprises two components: (i) an external notebook that stores compact, task-relevant notes across the entire episode, and (ii) a deletion action that can remove the selected past interactions from the context. This design is inspired by the Pensieve metaphor in *Harry Potter*, where raw experiences are extracted, distilled into durable memories, and revisited on demand.

Concretely, in addition to standard reasoning and tool-use actions, StateLM may emit context-management actions of the form

$$a_t = \texttt{note}(args), \quad a_t = \texttt{deleteContext}(msg),$$

where $msg$ identifies a previous assistant action $a_i$ or environment observation $o_i$. Executing `deleteContext()` removes the corresponding element from the interaction history exposed to the model in subsequent rounds, while the recorded note remains persistently accessible. This helps to sustain long-horizon reasoning under fixed context budgets.

### 3.3 TRAINING APPROACH

#### 3.3.1 SUPERVISED LEARNING FROM EXPERT TRAJECTORIES

We first initialize StateLM using supervised learning from expert-generated trajectories that demonstrate effective context management. To construct such trajectories, we incorporate a teacher model and place it in the StateLM inference environment, equipped with a curated system prompt that specifies high-level process guidelines and a few processing examples. We then collect complete interaction trajectories produced by the teacher. As illustrated in Figure 3 (left), each expert trajectory consists of a sequence of interleaved actions, decomposed into thoughts ($t$) and acts ($a$), together with the corresponding environment observations ($o$).

In practice, trajectories produced by teacher models can fail to meet our requirements due to various reasons. To obtain a high-quality dataset, we apply a post-processing pipeline to filter and construct the training samples.

**Outcome-based Reject Sampling.** We first retain only trajectories that reach a correct final answer. Correctness is determined by comparing the model-provided answer against the golden answer.

**Process-based Reject Sampling.** Among the trajectories that pass outcome filtering, we further filter based on context management behavior. Specifically, we discard trajectories that fail to prune the context in time, and those that skip necessary reading steps when sequential reading is required.

**Training Sample Construction.** From each expert trajectory $T = (q, t_1, a_1, o_1, \ldots, a_T)$, we construct multiple supervised training samples by progressively evolving the interaction history within the same StateLM environment used during generation. Specifically, at the $i$-th step, the input context to the model is the chat-templated text derived from the interaction state $s_i$, which recursively evolves from the initial state $s_0$, while the prediction target is the teacher's thought and act at step $i$.

During training, we apply a token-level mask such that the cross-entropy loss is computed only over the tokens of the final assistant turn corresponding to step $i$. All earlier turns are masked out and do not contribute to the loss, as they may be modified or removed during subsequent generation. This masking scheme also ensures compatibility with chat templates that include generation tag control, such as the `<think>` tag logic used by Qwen3.

**Action Balancing.** Even after the two-stage rejection sampling, the resulting data distribution can remain imbalanced. In particular, actions involving frequent and low-complexity tool calls such as `deleteContext` and `readChunk` may dominate the dataset, leading to the underrepresentation of other operations. To avoid biasing the policy toward certain operation strategies, we downsample training samples associated with overrepresented actions, yielding a more balanced distribution over memory and reasoning operations.

#### 3.3.2 REINFORCEMENT LEARNING FOR SELF-IMPROVEMENT

Following behavior initialization through supervised fine-tuning, we employ reinforcement learning fine-tuning to further improve StateLMs and promote the emergence of effective problem-solving strategies through trial-and-error interaction. The training algorithm builds on a GRPO-style objective, with adaptations specific to StateLM, including trajectory snapshots, controlled batching, and task-aware reward design.

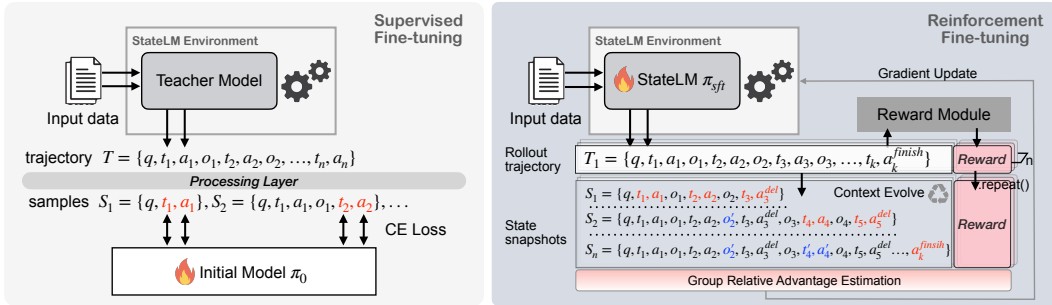

Figure 3: The two training stages of StateLM.

**Trajectory Rollout and State Snapshots.** As illustrated on the Figure 3 (right), given an input query $q$, we rollout $n$ trajectories $T^{(1)}, \ldots, T^{(n)}$ from the current policy $\pi_{\theta_{\text{old}}}$. Each trajectory corresponds to a complete episode and terminates either with a `finish` call or upon reaching a predefined budget limit. During rollout generation, the agent interacts with the environment and invokes context management actions. We collect a state snapshot whenever a *context-editing* action is taken. After the snapshot, the context is evolved and masked, and next sample proceeds from the new state. By the end of a trajectory, this process yields a sequence of state snapshots together with their loss masks.

Rather than using all snapshots produced within a trajectory, we control the batch size by uniformly sampling a fixed number of snapshots per trajectory. Each selected snapshot contributes equally to the optimization objective, independent of the length of the underlying trajectory. Under this design, the parameter $k$ serves as a control knob for variance and computational cost, thereby mitigating bias toward longer trajectories.

**Reward Design.** After collecting the rollout trajectories, we assign a scalar reward to each trajectory and its associated snapshot trajectories based on the outcome of the final trajectory. Specifically, given a query $q$, a golden answer $a$, and the terminal trajectory $T_{\text{last}}$, the reward is defined as:

$$R(q, a, T_{\text{last}}) = \begin{cases} +1, & \text{correct, formatted, and finished,} \\ -0.5, & \text{incorrect, formatted, and finished,} \\ -1, & \text{unformatted or unfinished.} \end{cases}$$

Correctness is determined by checking equivalence between the final answer and the golden answer using the same judging procedure as in the SFT stage. Format compliance requires the answer to be returned via the `finish` tool call and to satisfy task-specific constraints, such as brevity requirements. We additionally impose predefined limits on the maximum context window size and the number of interaction turns. Trajectories that exceed either limit are aborted and categorized as unfinished. Given this outcome reward, the advantage is computed using a group-based baseline formed across samples from multiple rollout trajectories.

## 4 EXPERIMENT

To verify the effectiveness of the proposed method, we evaluate the proposed method through extensive experiments on three base models of varying parameter scales across multiple benchmarks.

### 4.1 TRAINING SETUP

Our training procedure consists of two stages. For Stage 1: Supervised Learning from Expert Trajectories, we use the PublicDomain split of NovelQA (Wang et al., 2025) and the training split of NarrativeQA (Kočiský et al., 2018) to generate expert demonstrations. These datasets are chosen for their long-context characteristics, which naturally require explicit context management. We use Claude Opus 4.1 as the teacher model for its agentic capabilities. While the majority of trajectories

Table 2: Needle-in-a-haystack (NIAH) experimental results comparing model performance across selected context lengths. All values represent accuracy (%) and results are averaged over 3 runs.

| Model | Length | | | | | | | |
|---|---|---|---|---|---|---|---|---|
| | 32K | 64K | 128K | 256K | 512K | 768K | 1M | 2M |
| Qwen3-4B | **100.00** | **100.00** | 88.33 | 41.67 | 23.33 | 16.67 | 3.33 | 1.7 |
| **StateLM-4B** (w/o search) | 95.00 | 95.56 | **95.56** | **88.33** | **76.67** | **62.78** | **53.89** | **32.22** |
| Qwen3-8B | **100.00** | **100.00** | 88.33 | 41.67 | 23.33 | 16.67 | 3.33 | 1.7 |
| **StateLM-8B** (w/o search) | 99.44 | 98.33 | **98.33** | **99.44** | **95.55** | **88.89** | **88.89** | **67.22** |
| Qwen3-14B | **100.00** | **100.00** | 88.33 | 41.67 | 23.33 | 16.67 | 3.33 | 1.7 |
| **StateLM-14B** (w/o search) | **100.00** | **100.00** | **99.44** | **97.78** | **89.45** | **94.44** | **95.00** | **83.89** |

are generated with the search tool enabled, we also incorporate a small subset generated without search to develop scan-based reading behaviors. In total, we collect 3.3K full trajectories, which are subsequently filtered and decomposed into 35.7K training samples. For Stage 2: Reinforcement Learning for Self-Improvement, we adopt LongBench v2 (Bai et al., 2025) as the training dataset, using 488 problems in its training set. We also augment this dataset by converting a subset of multiple-choice questions into open-ended questions based on LLM judgments. The detailed statistics for the training datasets are described in Appendix A, and the training configurations used in the experiments are reported in Appendix B.

For the models, we conduct experiments using the 4B, 8B, and 14B instruct variants from Qwen3 model family (Yang et al., 2025), selected for their strong capability and agentic support. Models trained with SFT and RL are referred to as **StateLM** and **StateLM-RL** in the following evaluations.

## 4.2 SYNTHETIC MEMORY RETRIEVAL

We first assess whether StateLM can succeed on a task that isolates memory retrieval from extensive reasoning. To this end, we follow the **Needle-in-a-Haystack** setup from prior work (Hsieh et al., 2024) and construct a synthetic benchmark of 480 problems spanning 8 context lengths. Each problem embeds a single key sentence (the "needle") into a long passage (the "haystack"), and the model must retrieve the exact value associated with the queried key. To test the model limit, we utilize YaRN (Peng et al., 2023) to obtain the recommended 128K context window for both StateLM and the Qwen3 instruct baselines. For the latter, we follow prior work (Wang et al., 2025; An et al., 2024) and truncate the input by retaining only the first 128K tokens of the context.

To test StateLM's memory management ability under increasing context pressure, we deliberately **remove** the search tool from the available tool set. Otherwise, StateLM would rely on the search tool to retrieve the key and achieve near-perfect accuracy regardless of context length. In addition, we adopt a context-deletion strategy in which only the assistant's tool-call segment is pruned when the corresponding assistant message ID is specified for deletion, while the original thought is preserved. We find empirically that this strategy better aligns the model's behavior with the scanning-based tool set and consistently yields higher performance.

Table 2 reports the results. Across all model scales, the StateLM variants outperform their instruct baselines starting from 128K and on. The Qwen3 instruct models report identical accuracy as they consistently solve the problems within their context limit, and repeated sampling does not affect the outcome. While the instruct baselines rapidly degrade beyond the 128K limit, falling to nearly 0% accuracy at 1M tokens, the StateLM variants remain highly robust up to 2M context.

## 4.3 LONG-CONTEXT REASONING

We next evaluate the effectiveness of StateLM in practical long-context reasoning scenarios. To this end, we consider four benchmarks spanning three different domains:

- **Long Document QA**: the Copyright split of NovelQA (Wang et al., 2025) (757 problems) and the En.MC (English Multiple-Choice) split of ∞Bench (Zhang et al., 2024) (229 problems).

Table 3: Performance comparison of StateLM with baseline models on long-context reasoning benchmarks. The mean input length of each benchmark computed by the Qwen3 tokenizer is reported in parentheses. Results for Qwen3 and StateLM variants are measured over 3 runs. *For BrowseComp-Plus, we use the full 128k context for our trained StateLMs for better performance.

| Model | Context | LongDoc QA (135K) | | Chat Memory (115K) | *BrowseComp+ (552K) |
| | | NovelQA | ∞Bench | | |
| --- | --- | --- | --- | --- | --- |
| Qwen3-235B (w/ Pensieve) | 256K | 80.71 | 73.36 | 67.00 | 55.33 |
| RL-MemoryAgent-7B | 32K | 60.24 | 62.45 | 40.60 | - |
| RL-MemoryAgent-14B | 32K | 78.86 | 74.24 | 59.00 | - |
| ReadAgent-8B | 32K | 16.38 | 24.02 | 0.00 | - |
| ReadAgent-14B | 32K | 23.12 | 34.06 | 14.60 | - |
| Qwen3-4B | 128K | $65.17 \pm 0.53$ | $59.97 \pm 0.50$ | $39.53 \pm 1.36$ | $2.89 \pm 1.02$ |
| **StateLM-4B** | 32K | $\mathbf{79.57} \pm 0.93$ | $\mathbf{67.25} \pm 0.76$ | $\mathbf{59.33} \pm 0.23$ | $\mathbf{35.33} \pm 5.92$ |
| Qwen3-8B | 128K | $65.87 \pm 1.42$ | $66.81 \pm 1.16$ | $45.40 \pm 1.56$ | $5.56 \pm 0.77$ |
| **StateLM-8B** | 32K | $83.84 \pm 0.42$ | $70.16 \pm 1.33$ | $58.93 \pm 2.53$ | $46.22 \pm 1.68$ |
| ↳*StateLM-8B-RL* | 32K | $\mathbf{84.15} \pm 1.00$ | $\mathbf{73.07} \pm 1.33$ | $\mathbf{59.73} \pm 2.20$ | $\mathbf{46.44} \pm 0.77$ |
| Qwen3-14B | 128K | $77.94 \pm 0.26$ | $74.96 \pm 0.25$ | $54.07 \pm 0.76$ | $5.46 \pm 0.04$ |
| **StateLM-14B** | 32K | $84.15 \pm 0.82$ | $77.44 \pm 1.40$ | $64.40 \pm 1.40$ | $51.33 \pm 1.34$ |
| ↳*StateLM-14B-RL* | 32K | $\mathbf{84.85} \pm 0.42$ | $\mathbf{78.46} \pm 0.67$ | $\mathbf{64.47} \pm 0.50$ | $\mathbf{52.67} \pm 4.00$ |

- **Chat Memory**: LongMemEval-S (Wu et al., 2024) (500 problems), which evaluates long-term memory retention under multi-turn user-assistant interactions.
- **Deep Research**: a randomly downsampled subset of BrowseComp-Plus (Chen et al., 2025) (150 problems) following (Sun et al., 2025a), which tests agent model's deep research ability through iterative reasoning over search results.

Our primary comparisons are against Qwen3 instruct models, which are scaled to 128K context length using YaRN. In addition, we include the following methods:

- **Qwen3-235B** (Yang et al., 2025): the strong agentic model Qwen3-235B-A30B-Instruct-2507 directly equipped with our StateLM framework with 256K context window.
- **ReadAgent** (Lee et al., 2024): a prompting-based LLM agent that organizes long documents into compressed episodic "gist" memories and selectively revisiting the original text when needed.
- **MemAgent** (Yu et al., 2025): an RL-trained LLM agent that processes long contexts incrementally in chunks and updates its memory prompt after each chunk.

For all benchmarks, we report **Accuracy** as the evaluation metric. For Long Document QA, where all questions are multiple-choice, correctness is determined using a rule-based evaluation script. For Chat Memory and BrowseComp-Plus, we follow the original evaluation protocols and adopt an LLM-based judge. Additional evaluation details and implementations are provided in Appendix C.

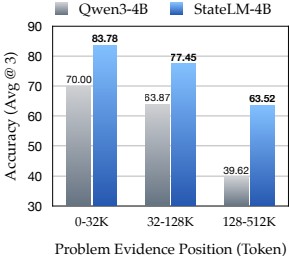 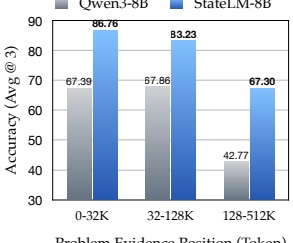 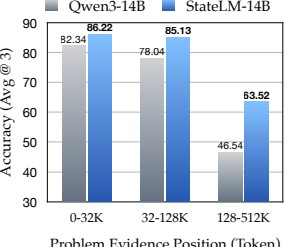

Figure 4: NovelQA accuracy by answer evidence token position in the provided context.

**On LongDocQA and Chat Memory, StateLM outperform the instruct baselines using only 1/4 of the context window and other agentic methods under the same context budget.** The results in

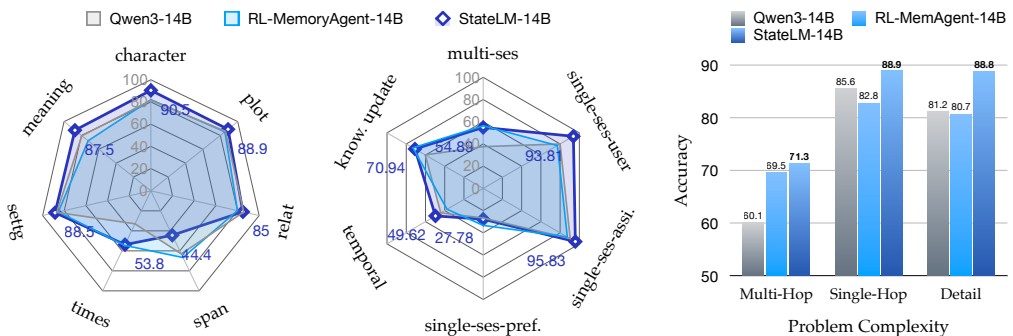

Figure 5: Performance breakdown by problem aspect on NovelQA (left) and LongMemEval (middle), and by question complexity on NovelQA (right).

Table 3 show notable performance improvements of StateLM over the standard instruct baselines, demonstrating the effectiveness of learned context management. For example, compared to the standard Qwen3-8B, StateLM-8B achieves over a 10% absolute improvement on Long Document QA, 13% on Chat Memory. While the relative gains are more pronounced for smaller models, StateLM follows a similar scaling trend to instruct models, with larger variants achieving stronger performance, indicating that the proposed approach scales effectively with model capacity. In addition, StateLM consistently outperforms other agentic memory methods at the same model scale and context budget, highlighting the advantage of the self-managed paradigm over predefined workflows.

Figure 4 further breaks down performance by the position of answer evidence in the context. The results show that, on the Long Document QA task, StateLM consistently outperforms the instruct baseline across all evidence ranges, with the most pronounced gains occurring when the relevant evidence appears later in the document. For example, in the 128–256K range, StateLM exceeds the instruct model by 23 and 24 accuracy points for the 4B and 8B variants, respectively.

**StateLM generalizes effectively to deep research tasks.** On the deep research benchmark BrowseComp-Plus, the performance gap becomes even more pronounced: StateLM-14B-RL achieves up to 52% accuracy, whereas the best vanilla counterpart, Qwen3-14B, remains around 5%. On average, the StateLM series delivers improvements over 40 points over standard LLMs. These results demonstrate the generalizability of our proposed paradigm to diverse scenarios, including complex agentic tasks.

**RL training brings further improvements.** Table 3 also shows that applying reinforcement learning on top of a well-trained StateLM can further improve performance. For example, StateLM-8B-RL achieves an additional 3-point increase on ∞Bench. Our in-house experiments show that continuing SFT training with additional epochs can degrade performance, whereas RL training does not suffer from this issue. These results indicate that StateLM can further refine its capabilities by exploring and learning from its own trajectories.

**StateLM improves across problem aspects and difficulty levels.** By breaking down performance across different axes in Figure 5, we observe that StateLM-14B achieves consistent improvements over both the instruct baseline and MemAgent across most aspects. StateLM is relatively weaker on the "span" and "user-preference" categories, which likely stems from the limitations of keyword-search-based reading when handling non-direct or implicit queries. In contrast, StateLM-14B exhibits the largest gains over the instruct model on the "Multi-Hop" category, showing the advantage and robustness of the proposed iterative reasoning loop for integrating evidence across multiple locations.

## 5 FURTHER ANALYSIS

### 5.1 TOOL USE PATTERN

To better understand how StateLMs manage their reasoning states during inference, we analyze their tool-use patterns across different benchmarks. Specifically, we track the total rounds, external memory operations (**mem**, including `note`, `updateNote`, and `readNote`), context pruning operations (**del**, `deleteContext`), and search operations (**srh**, `searchEngine`). Table 4 reports

the average counts of these operations per problem for StateLM-14B, and more results are provided in Appendix D.

The results reveal several patterns. First, the total number of reasoning rounds increases with task difficulty. In particular, StateLM allocates more intermediate steps on more challenging benchmarks such as LongMemEval and BrowseComp-Plus, where the overall performance is lower. Second, as input length grows, the primary change is an increase in search operations rather than memory updates, suggesting that StateLM relies on multiple targeted searches

|  | Rounds | mem | del | srh |
|---|---|---|---|---|
| NovelQA (119K) | 18.6 | 4.3 | 6.3 | 1.8 |
| ∞Bench (189K) | 20.7 | 4.6 | 6.8 | 2.9 |
| LongMemEval (115K) | 22.4 | 4.9 | 8.0 | 2.2 |
| BrowseComp+ (552K) | 22.8 | 4.1 | 6.1 | 6.6 |

Table 4: Tool-use pattern of StateLM-14B across benchmarks with mean input length reported.

to navigate longer inputs while retaining only key information in memory. Finally, the number of memory operations remains cost-efficient. In contrast to fixed-workflow methods where memory updates may occur at every step, StateLM invokes memory operations far less frequently. Together, these patterns demonstrate StateLM's ability to adapt its behavior to different tasks in an efficient and problem-aware manner.

## 5.2 COMPARISON TO QWEN3 AGENTIC

Since recent open-source models such as Qwen3 already support agentic tool calls, a natural question arises: is deliberate training necessary for developing StateLMs? In other words, can instruct models learn to manage their own context and memory solely through an agent-style system prompt paired with an appropriate toolkit? To validate our approach, we construct a detailed system prompt that explicitly specifies the intended context management procedure (see Figure 9 in the Appendix) and expose the full tool set of StateLM to Qwen3 instruct models, which we refer to as "Agentic" models. We then evaluate these models on the Long Document QA tasks including NovelQA and ∞Bench and compare their performance against our fine-tuned StateLMs.

The results in Figure 6 present this comparison. While the largest agentic model, Qwen3-14B, is able to correctly complete and submit answers for up to 30% of the questions, its performance remains substantially below that of the trained StateLMs. Smaller agentic variants perform even worse, further widening the gap. Our error analysis reveals that these models often fail to keep their context within the budget, causing the overflow error. These findings suggest that although such models inherently possess instruction-following and tool-calling capabilities, mastering the context management with the Pensieve is not a trivial task and requires deliberate training, particularly for these relatively small-scale models. In other words, our

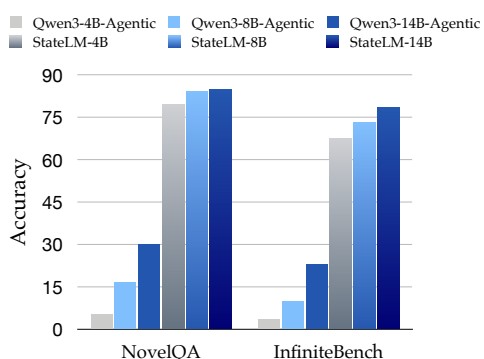

Figure 6: StateLM and Qwen3-agentic model performance comparison.

training stages enable these models to learn and develop effective tool-use patterns that make context management more robust and reliable.

## 6 CONCLUSION

In this work, we introduce StateLM, a state-aware agentic system in which models actively manage their own context through tool-based operations. Across both synthetic memory retrieval and realistic long-context reasoning tasks, StateLM consistently outperforms instruct baselines as well as existing agentic methods. These results point to a shift from systems that passively accumulate context toward models that dynamically refine and control their reasoning states. Despite certain limitations, our proposed paradigm offers a scalable and principled approach for future foundation models and agentic systems, where context management becomes an intrinsic and learned capability.

## ACKNOWLEDGEMENT

This paper was supported by the Guangdong Basic and Applied Basic Research Foundation (No. 2024A1515010145).

## REPRODUCIBILITY STATEMENT

We ensure reproducibility by providing detailed descriptions of our methodology, datasets, models, and training and evaluation configurations in Section 4, Appendix B, and Appendix C. The main code is included in the supplementary material, and we will release both the code and our training dataset publicly to facilitate the reproduction of our results.

## ETHICS STATEMENT

This work complies with the ICLR Code of Ethics. Our research mainly develops a stateful language model that can efficiently manage its own context memory. Training and evaluation rely solely on published datasets, and the research does not involve human subjects or sensitive personal data. We carefully considered issues of fairness, privacy, security, and potential societal impact, and found no outstanding ethical concerns. Nevertheless, we acknowledge that such models are still at an early stage of development and should be used for research purposes only. We further emphasize that evaluations should be conducted in controlled environments with limited tool access.

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

## THE USE OF LARGE LANGUAGE MODELS

Large language models (LLMs) were applied in a restricted manner to support writing refinement. The authors supplied their draft text to the LLM, which suggested enhancements such as grammatical corrections, clearer formulations, and the elimination of informal expressions. LLMs were additionally consulted to generate possible paper titles. While the model proposed alternatives, the final title was independently selected and refined by the authors, and was not adopted verbatim from any single model output. Furthermore, LLMs were employed as auxiliary coding tools during implementation. They provided code completions and debugging advice; however, all final code, experimental setup, and validation were designed, implemented, and confirmed by the authors. It is important to emphasize that LLMs **WERE NOT** utilized for developing research ideas, designing experiments, or conducting literature review. All conceptual advances and experimental methodologies were entirely conceived and executed by the authors.

## A  TRAINING DATASET CONSTRUCTION

| Source | Questions | Trajs | Outcome Filter | Process Filter | Samples | Action Balancing |
|--------|-----------|-------|----------------|----------------|---------|------------------|
| NovelQA | 3096 | 3291 | 2322 | 2256 | 46180 | 35737 |
| NarrativeQA | 100 | 83 | 8 | 8 | 634 | |
| Total | 3196 | 3374 | 2330 | 2264 | 46814 | 35737 |

Table 5: SFT Dataset statistics by filtering stages.

| Source | Questions | Type | Split |
|--------|-----------|------|-------|
| LongBench v2 | 488 | Open-Ended (127), MCQ (361) | Train |
| | 15 | Open-Ended (1), MCQ (14) | Validation |

Table 6: RL Dataset statistics.

For LongBench v2 data used in reinforcement learning, we convert a subset of multiple-choice questions into open-ended questions by removing the original option lists, which help to diversify the problem type. The transferability of the question is determined by an LLM-based judge, which we instantiate as gpt-oss-120b, based on whether the correct answer remains unique, verifiable, and unambiguous.

## B  TRAINING CONFIGURATIONS

For Supervised Fine-tuning (SFT), each model was trained on two machines, each equipped with 8 H20 GPUs. The training configurations are specified in the table below.

Table 7: SFT Training Configuration

| Parameter | Value |
|-----------|-------|
| Global Batch Size | 128 |
| Learning Rate Scheduler | cosine |
| Learning Rate | $1 \times 10^{-5}$ |
| Warm-up Ratio | $3 \times 10^{-2}$ |
| Max Sequence Length | 28000 |
| Epochs | 3 (14B), 4 (4B, 8B) |
| Parallelism Strategy | DeepSpeed ZeRO-3 |

For reinforcement learning (RL) training, we use the GRPO algorithm with a rollout batch size of 32 and a rollout number of 8. We enable KL loss and set the KL coefficient to 0.001. The number of

samples for each rollout trajectory is set to 8 for the 8B model and 2 for the 14B model. Models are trained for 32 steps to obtain the StateLM-RL variants.

## C   EVALUATION CONFIGURATIONS

### C.1   GENERATION

In our experiments, we use the Qwen3 non-thinking mode for both StateLMs and instruct baselines. We exclude the thinking mode because it requires a 32K output space at each step, which is computationally expensive and reduces the available context window.

For both StateLM and instruct baselines, we adopt the recommended sampling parameters as documented in the Qwen3 official guide. Detailed configurations are reported in Table 8 below:

Table 8: Generation Configuration

| Parameter | Value | Method |
|---|---|---|
| Temperature | 0.7 | Both |
| Top_p | 0.8 | Both |
| Top_k | 20 | Both |
| Max Output Tokens | 8000 | Instruct |
| Max Context Tokens | 120000 | Instruct |
| Context Budget | 32000 | StateLM |
| Round Budget | 150 | StateLM |
| Max Rounds | 200 | StateLM |

This configuration is used for most evaluations, except for the BrowseComp-Plus subset. For this benchmark, we follow the official configuration to obtain the Qwen3 results, as detailed in the official repository.[1] In this setting, Qwen3 instruct models are augmented with a search tool based on the BM25 algorithm, which is the same algorithm used in StateLM's search tool.

### C.2   GRADING

For long-document QA tasks with multiple-choice questions, we use a rule-based grading script adapted from the implementation of (Zhang et al., 2024). Additionally, we observe that MemAgent models often output the option content *without* the corresponding option letter in multiple-choice questions; to reflect their best achievable performance, we adapt our grading script to treat such responses as correct. For LongMemEval and BrowseComp-Plus, we follow the official evaluation guidelines and use an LLM-based judge to assess answer correctness, with GPT-4o as the judge model.

For the reward module in RL training, we adopt a two-layer design: multiple-choice questions are evaluated using a rule-based grading script, and open-ended questions are evaluated with an LLM-based judge. In the experiment, the judge model is set to gpt-oss-120b, and the grading prompt is shown in Figure 8.

## D   TOOL USE ANALYSIS

In this section, we report tool-use statistics for StateLM-4B and StateLM-8B, which complement the analysis presented in Section 5.1.

---

[1] `https://github.com/texttron/BrowseComp-Plus/blob/main/docs/qwen.md`

Table 9: Tool-use pattern of StateLM-4B across benchmarks. Mean input length of each benchmark is reported.

|  | Rounds | mem | del | srh |
| --- | --- | --- | --- | --- |
| NovelQA (119K) | 19.5 | 4.4 | 6.5 | 2.3 |
| ∞Bench (189K) | 21.4 | 4.6 | 6.6 | 3.9 |
| LongMemEval (115K) | 24.5 | 5.3 | 8.3 | 3.5 |
| BrowseComp+ (552K) | 21.9 | 3.4 | 5.4 | 7.2 |

Table 10: Tool-use pattern of StateLM-8B across benchmarks. Mean input length of each benchmark is reported.

|  | Rounds | mem | del | srh |
| --- | --- | --- | --- | --- |
| NovelQA (119K) | 19.6 | 4.5 | 6.6 | 2.2 |
| ∞Bench (189K) | 21.0 | 4.3 | 6.3 | 4.2 |
| LongMemEval (115K) | 24.5 | 5.6 | 8.9 | 2.3 |
| BrowseComp+ (552K) | 22.5 | 3.8 | 5.2 | 7.9 |

## E    ERROR ANALYSIS

To better understand the failure modes of StateLM, we conduct a qualitative error analysis by manually inspecting incorrect trajectories across the evaluation benchmarks, and the major failure modes are summarized as follows:

- **Search Constraints**. The current BM25-based keyword retrieval often misses evidence for implicit or paraphrased queries due to limited semantic coverage.
- **Formatting Errors**. A small fraction of failures arise from malformed tool calls, particularly in long-horizon trajectories and smaller models.
- **Context Window**. Deleted content is replaced by lightweight stubs, which can still accumulate over long trajectories and gradually consume context budget. In addition, untimely or overly conservative deletions may cause transient context overflow before pruning takes effect.

These failure modes suggest several promising directions for StateLM improvement, including larger management windows, more advanced search algorithm, and more high-quality on long-horizon training trajectories, which we leave for future work.

## F    PROMPTS

> **Prompt**
>
> You are an AI assistant for long-context processing with tools. Produce factually correct answers grounded in any attached text while conserving the context window by deleting unnecessary messages and taking notes. Describe your processing plan first, then proceed with the tools.

Figure 7: System prompt used in training and inference of our StateLMs.

Prompt

Given a problem, its correct answer, and a student's answer below, your task is to review the student's answer and determine if it is correct by comparing it to the correct answer. If the student's answer is incomplete or ambiguous, assume it is incorrect.

### Problem
{problem}

### Answer
{answer}

### Student Answer
{mode_ans}

Please put your final answer (True or False) in \\boxed{}. Specifically, if the student's answer is correct, the final answer should be \\boxed{True}; otherwise, the final answer should be \\boxed{False}.

Figure 8: Prompt template used by LLM Judge for Open-Ended questions.

Prompt

You are an AI assistant specialized in processing long-context tasks with tools. Produce factually accurate answers grounded in the provided context while minimizing context consumption.

Processing Strategy:
1. Check the size of the attached text:
   - Long ($> 8K$ tokens): build an index and process in chunks. For extremely long texts, increase the chunk size up to 12,000 tokens.
   - Short ($\leq 8K$ tokens): load the full text and answer directly.
   - Empty: proceed with reasoning, using note-taking tools.
2. Analyze user's query and justify which processing mode is required to answer reliably and state that you plan to use that mode explicitly.
   (a) Linear scan: Full-passage, sequential chunk-by-chunk reading (no details skipped), or
   (b) Keyword search: Keyword-based search to retrieve and inspect only the relevant chunks.
3. While reading, record relevant, accurate, and verifiable notes. Merge related notes as they grow to keep them concise.
4. Delete unnecessary context messages by their 'msg_id' to preserve context space, but do not delete everything or overuse the deletion tool. Deleted messages become stubs-do NOT restate their contents. Two required cases for deletions:
   - After calling 'readChunk': once you have analyzed the chunk and optionally taken notes, immediately delete the chunk content using the 'msg_id' returned by the 'readChunk' tool.
   - After calling 'note': delete the invoking assistant message using the 'msg_id(invoking_assistant)' returned by the 'note' tool result so the note-construction message is cleared.
5. Consult your notes and use relevant evidence to answer the user's query.
6. Call 'checkBudget' regularly to monitor usage and prevent overflows; adjust your strategy accordingly.

Describe your reasoning and processing plan before invoking any tools.

Figure 9: System prompt for the Qwen3-Agentic models.

