# OpenReview forum: "The Pensieve Paradigm: Stateful Language Models Mastering Their Own Context"
_ICLR.cc/2026/Conference — ICLR 2026 Poster_

### Official Review · Reviewer_xaus · 2025-10-28

**Soundness:** 3
**Presentation:** 3
**Contribution:** 3
**Rating:** 6
**Confidence:** 4

**Summary:**

The paper introduces **The Pensieve Paradigm**, a framework that equips an LLM with tools to dynamically manage its own context and reasoning process. The proposed system, called **StateLM**, allows the model to actively construct, prune, and update its working memory through operations such as dynamic indexing, note-taking, and context deletion. This enables it to go beyond the predefined context length and handle extremely long sequences efficiently.

The authors propose a data generation pipeline, and employ SFT to train models from the Qwen3 family. Empirical results show that StateLM achieves strong performance in both synthetic and real-world tasks, consistently outperforming the baselines, while using only a fraction of the context length.

**Strengths:**

- The paper is well written, and the main ideas are presented clearly.
- I believe that the problem that the authors tackle is both interesting, and useful, particularly in the context of agentic systems. In many cases, agents are required to handle huge contexts (for instance large code repos); therefore, the proposed paradigm could be leveraged to both improve the performance and make agentic systems more efficient.
- The paper has solid amount of empirical results that showcase the strength of the proposed method, covering both synthetic and real-world settings.

**Weaknesses:**

- As the authors also mention, the current set of tools is predefined and fixed. While this appears sufficient for the evaluated tasks (Needle-in-a-Haystack and Long Document QA), it may prove inadequate in more complex or dynamic scenarios
- In Section 4.1, I think that the comparison with the baseline may not be entirely fair. The high accuracy achieved by the proposed approach on extremely long contexts is indeed interesting;  however, the poor performance of the baseline models in the cases where the context length is exceeded is not surprising. Perhaps, a better baseline could be to use a simple sliding window approach, where the size of the window is close to the context size of the model.
- Although the authors note this as well, I believe that testing RL approaches would be reasonable in this setting, since there has been evidence that RL works well in similar settings(e.g [1]).

[1] Feng, Jiazhan, et al. "Retool: Reinforcement learning for strategic tool use in llms." arXiv preprint arXiv:2504.11536 (2025).

**Questions:**

- In Figure 5, why does the inference time of  Qwen3-8B (baseline) decrease as the context length increases? Is it due to truncation?

---

> ### Author Response · Authors · 2025-11-21
> **Response to Reviewer xaus (1)**
>
> We thank the reviewer for the positive assessment of our work by finding our target problem "interesting, and useful",  and the empirical results are strong. Below we address the raised concerns in detail.
>
> > **[W1] Fixed toolset limitations.**
> >
>
> To use an analogy fitting our paper's theme: In the current iteration of the *Pensieve Paradigm*, our model acts as a **junior wizard**. It is learning to proficiently wield a set of "textbook spells" (our pre-defined tools) to manipulate memory. While this "curriculum" is sufficient to validate the effectiveness of the paradigm on current benchmarks, we acknowledge that dealing with highly dynamic real-world scenarios will eventually require **"true wizardry",** enabling the model to **invent its own spells** (dynamically construct tools).
>
> We view this paper as establishing the **fundamental mechanics**: with a fixed toolset, the model can already learn to **dynamically** manipulate its own state. Once this foundation is solid, giving the model the autonomy to craft its own tools is the logical and necessary next step.
>
> **[W2] Fairness of baselines and sliding-window alternatives.**
>
> We appreciate the reviewer’s concern regarding the baseline choice. Our evaluation follows established practice in long-context benchmarks [1,2], where models are assessed by truncating inputs to their maximum supported length. A “sliding-window” strategy would still face the same limitation: **once the input exceeds the model’s window, earlier tokens are inevitably omitted**, leading to the sharp performance degradation [3] that we also observe in our experiments.
>
> To alleviate this concern, we implemented MemAgent [4], which processes the document chunk-by-chunk and updates a fixed memory state after each chunk, closely mirroring the intended sliding-window heuristic. Our preliminary results (shown below) indicate that StateLM maintains substantially higher accuracy and greater stability across the evaluated benchmarks.
>
> |  | **NovelQA** | **InfiniteBench** |
> | --- | --- | --- |
> | RL-MemoryAgent-7B | 30.38 | 34.93 |
> | RL-MemoryAgent-14B | 39.50 | 45.85 |
> | **StateLM-4B (updated)** | 79.57 ± 0.53 | 69.58 ± 2.20 |
> | **StateLM-8B (updated)** | 83.71 ± 1.28 | 69.58 ± 1.01 |
> | **StateLM-14B (updated)** | 85.47 ± 0.82 | 77.29 ± 0.44 |
>
> Note that we have updated the search engine in our framework by improving the its frontend (tool results) with search previews and highlights, and upgrading the backend of using the more advanced ElasticSearch engine. We then retrained the models to obtain updated StateLMs. **You can also find the complete table of results in our general response.**
>
> [1] Wang, Cunxiang, et al. "NovelQA: Benchmarking Question Answering on Documents Exceeding 200K Tokens." *The Thirteenth International Conference on Learning Representations*.
>
> [2] An, Chenxin, et al. "L-eval: Instituting standardized evaluation for long context language models." *Proceedings of the 62nd Annual Meeting of the Association for Computational Linguistics (Volume 1: Long Papers)*. 2024.
>
> [3] Xiao, Guangxuan, et al. "Efficient Streaming Language Models with Attention Sinks." *The Twelfth International Conference on Learning Representations*.
>
> [4] Yu, Hongli, et al. "MemAgent: Reshaping Long-Context LLM with Multi-Conv RL-based Memory Agent." arXiv preprint arXiv:2507.02259 (2025).

---

> ### Author Response · Authors · 2025-11-21
> **Response to Reviewer xaus (2)**
>
> > **[W3] On reinforcement learning.**
> >
>
> We truly appreciate this insightful observation. This is also a fundamental question in training agentic systems: the trade-off between the stability of supervised synthesis and the autonomy of reinforcement learning.
>
> While we agree that RL represents an exciting direction, our choice to use a teacher-guided SFT pipeline was a deliberate strategic decision aimed at maximizing transferability and accessibility, which also aligns with widely adopted paradigms for training advanced reasoning models [1] and tool-use agents [2]. We would like to highlight two advantages of this approach:
>
> 1. **The Teacher is Flexible**: The "strength" of data generation stems less from the raw parameter scale and more from the sophisticated prompting coupled with our toolkit. This makes the pipeline accessible without strict reliance on SOTA closed-source models.
>
> 2. **Seamless Integration into Post-Training**:  the primary goal of this project is to evolve our foundation model itself, so we prioritized compatibility with standard post-training recipes. Our trajectories can be seamlessly mixed into general instruction-tuning datasets (e.g., alongside math or code) to endow foundation models with state-management capabilities. In contrast, RL requires complex reward engineering, making it difficult to integrate into the unified training pipelines of general-purpose LLMs.
>
> Nevertheless, we fully agree that RL is the natural next step to further extend this framework, and we are excited to explore it in future work.
>
> [1] Guo, Daya, et al. "Deepseek-r1: Incentivizing reasoning capability in llms via reinforcement learning." *arXiv preprint arXiv:2501.12948* (2025).
>
> [2] Qin, Yujia, et al. "ToolLLM: Facilitating Large Language Models to Master 16000+ Real-world APIs." *The Twelfth International Conference on Learning Representations*.
>
> > **[Q1] Why does the inference time of Qwen3-8B (baseline) decrease as the context length increases? Is it due to truncation?**
> >
>
> Yes, the drop in runtime is caused by truncation once the input exceeds 128K tokens. In these cases, the answer can be removed entirely from the input, making the problem essentially “easier” as the model hallucinates unsupported answers rather than performing grounded retrieval. Such hallucinated outputs introduces variability and can lead to slightly faster or more irregular inference time.
>
> ---
>
> Finally, we sincerely appreciate your constructive review and remain fully available to clarify any additional details during the discussion period. If our responses have addressed your concerns regarding toolset generalizability and baseline fairness, **we would be grateful for your support in the discussion stage**. We believe this work offers a meaningful step toward enabling LLMs to manage their own state, and your endorsement would be invaluable in helping bring these ideas to the community.

---

### Official Review · Reviewer_GVg6 · 2025-10-31

**Soundness:** 3
**Presentation:** 3
**Contribution:** 2
**Rating:** 4
**Confidence:** 4

**Summary:**

Pensieve/StateLM reframes long-context modeling as learned memory management. The LLM executes a tool-use loop ( e.g., analyzeText, buildIndex, searchEngine) to construct and prune its own working context rather than passively consuming a fixed window. StateLM is trained on Claude Sonnet 4–generated trajectories filtered for outcome and behavior on NovelQA and NarrativeQA. On Needle-in-a-Haystack (NIAH), StateLM outperforms Qwen-3 instruct baselines (4B/8B). On NovelQA and InfiniteBench, a 32K-context StateLM surpasses a 128K instruct baseline and shows scaling trends with higher inference-time compute. A prompt-only agent underperforms StateLM, supporting the claim that these behaviors should be learned, not merely prompted.

**Strengths:**

- The data curation pipeline is carefully designed. multi-stage filtering and process-mode classification (search vs. scan) produce cleaner trajectories for training.
- SLM w/o search greatly outperforms baseline by a large margin especially after 256K tokens.
- On real-world tasks like NovelQA and InfiniteBench, the results are impressive where SLM with short context (32K) can achieve better performance than instruct model with context of 128K token
- Good writing. The description of StateLM and experiments are clear and easy to follow.

**Weaknesses:**

- It is not clear how StateLM materially differs from prior work (e.g., A-Mem, SCM, Dynamic Cheatsheet). The claim of “not a fixed workflow loop” does not really establish novelty, as this function has been supported by agentic toolkit like Anthropic’s Model Context Protocol (MCP) and also has been explored by prior work.
- The training trajectories come from Sonnet-4, which along with many open-source agents already can decide which tools to use given context. As presented, the contribution is largely policy learning over a fixed toolset (index/search/read/note/delete), rather than a new memory paradigm.
- Ablations are limited: comparisons are mainly against a prompt-only baseline. There are no per-tool ablations (e.g., removing deleteContext) and no robustness analysis under noisy/failed tool calls.
- All results are based on only two models Qwen3 4B/8B of the same family. It is not clear if it generalizes well to other families.
- StateLM appears to use substantially more inference-time interaction/compute than single-pass baselines, making the comparison potentially not apples-to-apples.

References:

```
@article{wang2024openhands,
  title={Openhands: An open platform for ai software developers as generalist agents},
  author={Wang, Xingyao and Li, Boxuan and Song, Yufan and Xu, Frank F and Tang, Xiangru and Zhuge, Mingchen and Pan, Jiayi and Song, Yueqi and Li, Bowen and Singh, Jaskirat and others},
  journal={arXiv preprint arXiv:2407.16741},
  year={2024}
}
```

```
@article{xu2025mem,
  title={A-mem: Agentic memory for llm agents},
  author={Xu, Wujiang and Mei, Kai and Gao, Hang and Tan, Juntao and Liang, Zujie and Zhang, Yongfeng},
  journal={arXiv preprint arXiv:2502.12110},
  year={2025}
}
```

```
@article{yu2025memagent,
  title={MemAgent: Reshaping Long-Context LLM with Multi-Conv RL-based Memory Agent},
  author={Yu, Hongli and Chen, Tinghong and Feng, Jiangtao and Chen, Jiangjie and Dai, Weinan and Yu, Qiying and Zhang, Ya-Qin and Ma, Wei-Ying and Liu, Jingjing and Wang, Mingxuan and others},
  journal={arXiv preprint arXiv:2507.02259},
  year={2025}
}
```

```
@article{wang2023enhancing,
  title={Enhancing large language model with self-controlled memory framework},
  author={Wang, Bing and Liang, Xinnian and Yang, Jian and Huang, Hui and Wu, Shuangzhi and Wu, Peihao and Lu, Lu and Ma, Zejun and Li, Zhoujun},
  journal={arXiv preprint arXiv:2304.13343},
  year={2023}
}
```

**Questions:**

1. Do you report a baseline using Qwen3 + MCP tooling for different tasks? Since Qwen3 family natively support building AI agents with MCP protocols, the prompt-only ablation may underserve tool use capability of instruct model.
2. Do you provide error analysis for NIAH, NovelQA, and InfiniteBench for both StateLM and baselines (instruct and prompt-based)?
3. What is the impact of each tool (e.g., removing deleteContext) on accuracy? Can you provide further ablation analysis?

---

> ### Author Response · Authors · 2025-11-21
> **Response to GVg6 (1)**
>
> We thank the reviewer for the detailed feedback and for recognizing our data curation pipeline, empirical results, and clarity of presentation. Below we address each concern in detail.
>
> > **[W1] Novelty relative to prior work & [W2] Simple Policy learning over a fixed toolset**
> >
>
> We would like to revisit the Pensieve analogy to emphasize our core contribution.
>
> When Dumbledore’s mind becomes overburdened, he extracts memories into a Pensieve to revisit later. This act of “extraction” is the fundamental operation that allows the model to retain only the information needed for the next step, aligning directly with the objective of context engineering.
>
> To the best of our knowledge, **no prior work (including standard MCP agents) implements an operation comparable to `deleteContext`**. The closest example is our concurrent study, Memory-R1 [1], whose deletion mechanism acts on a separate memory bank rather than the active context itself. Thus, while our method may appear to be straightforward policy learning over a fixed toolset, we believe this is a case of **any mundane tool becoming a master's weapon**. The key outcome is that this design enables the model to learn to **manipulate its own state**.
>
> [1] Yan, Sikuan, et al. "Memory-r1: Enhancing large language model agents to manage and utilize memories via reinforcement learning." *arXiv preprint arXiv:2508.19828* (2025).
>
> > **[W3 & Q3] Tool-based Ablation and prompt-only baseline**
> >
>
> The prompt-only agent introduced in Section 4.4 is not a trivial, it is essentially a **function-calling variant with the full tool specification**, mirroring what MCP-style systems support. As shown in our ablation, this configuration remains far below StateLM across all datasets, indicating that the learned usage pattern, not the mere availability of tools, is the critical factor.
>
> The toolset is designed as a minimal and tightly-coupled system. Ablating on each tool is costly and does not provide useful information as in most cases removing any single tool fundamentally breaks the workflow: without `deleteContext` the model cannot maintain a stable active window, without `buildIndex` and`readChunk` it cannot access the document, and without `updateNote` it cannot accumulate evidence across rounds. Nevertheless, we report the average **tool frequencies** of StateLM-8B on NovelQA and InfiniteBench to reflect **each component’s contribution**. These statistics show that deleteContext is invoked consistently and is tightly aligned with successful reasoning trajectories, whereas auxiliary tools show varied but interpretable usage patterns.
>
> | **StateLM-8B** | **NovelQA** | **InfiniteBench** |
> | --- | --- | --- |
> | analyzeText | 1.0 | 1.0 |
> | buildIndex | 1.0 | 1.0 |
> | readChunk | 3.31 | 3.24 |
> | note | 1.15 | 1.29 |
> | deleteContext | 6.56 | 6.21 |
> | updateNote | 2.12 | 1.76 |
> | readNote | 1.19 | 1.27 |
> | searchEngine | 2.03 | 3.47 |
> | checkBudget | 0.01 | 0.05 |
> | finish | 0.99 | 0.93 |
>
> > **[W4] Generalization across models**
> >
>
> We add the experiment for **Qwen3-14B** model, which further improves the model scale. Note that we choose Qwen3 models mainly for their agentic capabilities, which lies the foundation for the StateLMs. You can also find the complete table in our general response.
>
> |  | NovelQA | InfiniteBench |
> | --- | --- | --- |
> | Qwen3-4B-128K | 65.17 ± 0.53 | 59.97 ± 0.50 |
> | **StateLM-4B (updated)** | **79.57 ± 0.53** | **69.58 ± 2.20** |
> | Qwen3-8B-128K | 65.87 ± 1.42 | 66.81 ± 1.16 |
> | **StateLM-8B (updated)** | **83.71 ± 1.28** | **69.58 ± 1.01** |
> | Qwen3-14B-128K | 77.94 ± 0.26 | 74.96 ± 0.25 |
> | **StateLM-14B (updated)** | **85.47 ± 0.82** | **77.29 ± 0.44** |
>
> Note that we have updated the search engine in our framework by improving the its frontend (tool results) with search previews and highlights, and upgrading the backend of using the more advanced ElasticSearch engine. We then trained the models to obtain updated StateLMs.
>
> > **[W5] Inference-time comparsions to baseline**
> >
>
> StateLM is intentionally designed to convert long input contexts into a sequence of inference-time interactions. Our measurements show that runtime grows linearly with input length while maintaining stable accuracy, in contrast to baselines that collapse once their fixed window is exceeded. This provides a practical way to address the architectural limits of current LLMs.
>
> > **[Q1] Report the baseline of Qwen3 + MCP tools**
> >
>
> Yes. In Section 4.4, our “prompt-only” baseline in Section 4.4 is implemented as **a function-calling agent with the full toolset**, effectively matching the capability of MCP-style tooling (MCP and function-calling differ only at the user side, while the model receives the same prompt structure). Its performance reflects the limitation of prompt-level tool orchestration without learning, which is consistently outperformed by the trained StateLMs.

---

> ### Author Response · Authors · 2025-11-21
> **Response to GVg6 (2)**
>
> > **[Q2] Error analysis for NIAH, NovelQA, InfiniteBench**
> >
>
> Thank you for the suggestion. We have summarized the typical error categories below. These analyses show that the primary failure modes of the instruct baselines stem from long-context hallucinations and truncation, whereas StateLM’s errors mainly arise from occasional tool misoperations and information accumulation over long reasoning chains. The detailed error analysis will be included in the appendix.
>
> |  | Qwen3 Instruct  | StateLM |
> | --- | --- | --- |
> | **NIAH Primary Errors** | (1) Truncated input leading to missing evidence (2) Long-context hallucinations, where the model observes the evidence but still answers incorrectly | (1) Note misoperations, such as overwriting correct notes or failing to read previously stored notes in time (2) API errors, including malformed tool calls or exceeding maximum output length |
> | **LongDoc QA Primary Errors**  (NovelQA & InfiniteBench)  | (1) Long-context hallucinations (2) Truncated input leading to missing evidence | (1) Search limitations, where keyword-based retrieval misses details not captured by search (2) Long-span information loss, where the model struggles to assemble multi-step evidence over extended reasoning (3) Untimely deletions that cause context overflow |
>
> ---
>
> We understand that your initial rating (4) was primarily driven by questions regarding **novelty relative to prior work** and the **comprehensiveness of baselines**. We have attached **expanded results in our general response;** In this response, we have systematically addressed these concerns:
>
> 1. **Established Core Novelty:** StateLM introduces a unique learned capability, **Active Context Pruning,** to enable true long-context sustainability. While the implementation utilizes policy learning over a fixed toolset, we view this as a case of **"a mundane tool becoming a master's weapon."** The critical contribution is the model's learned autonomy to manipulate its own cognitive state, rather than the complexity of the tools themselves.
> 2. **Expanded Evaluation:** We have added **new baselines** (Mem-Agent, ReadAgent), **broader benchmarks** (LongMemEval), and **larger model scales** (Qwen3-14B), all of which confirm the robustness of our approach.
>
> Given that we have directly addressed the specific concerns regarding novelty and experimental scope that grounded your initial assessment, we respectfully request that you reconsider your rating. We believe the revised paper offers a distinct and valuable contribution to our community!

---

> ### Comment · Reviewer_GVg6 · 2025-11-27
> **Response to Authors**
>
> We appreciate the follow-up experiments and clarifications. While the paper frames its core contribution as an operation enabling the model to extract or delete information from its context, we remain unconvinced about its novelty. The proposed mechanism reads primarily as an engineering feature for context management, presented through a Harry Potter analogy. Moreover, prior work such as MemGPT has already introduced model-invoked context eviction by allowing an LLM to “retrieve relevant historical data missing from what is placed in-context, and also evict less relevant data from context" in order to address the same limited context problem. The specific realization of deleteContext within a stateful LM appears incremental rather than fundamentally new. We therefore maintain the original score.
>
>
> References:
> - Packer, Charles, et al. "MemGPT: Towards LLMs as Operating Systems." (2023).

---

### Official Review · Reviewer_qeFc · 2025-11-01

**Soundness:** 3
**Presentation:** 3
**Contribution:** 2
**Rating:** 4
**Confidence:** 3

**Summary:**

This paper presents StateLM, a stateful, memory-augmented LLM agent supervisedly trained to
autonomously manage its own context through a (predefined) set of tools. Instead of relying on a
human-defined workflow, the model learns, via SFT on author-curated reasoning trajectories, to
decide when to perform indexing, note-taking, searching, and pruning. Experiments on long-
context QA benchmarks, compared with Qwen3-Instruct models, demonstrate substantial gains
in both efficiency and scalability.

**Strengths:**

● The problem addressed in this paper is crucial: transitioning from stateless LLMs to a
stateful paradigm enables long-term reasoning, multi-turn dialogue memory, and
cross-session continuity.
● The paper is well-written, and the case study in Section 3 provides an intuitive and
effective way to illustrate the Pensieve paradigm.
● The “model as the wizard” framing, i.e., pushing the model toward fully autonomous
decision-making about when (and, potentially in future work, how) to manage its own
context, is an appealing selling point that makes the work conceptually engaging.
However, as discussed below in the weaknesses, it remains to be seen whether this
setting can match the performance of prior semi-automated workflows.

**Weaknesses:**

1. The paper aims at a meaningful goal of achieving a fully automated workflow, since
heuristic and human-defined pipelines may not fully unlock the capability of LLMs.
However, the framework still relies on manually defined tools, making it essentially
semi-automated. Given that prior work (e.g., Memory-R1) also trains models to learn
what memory operations to perform, the main difference here seems to lie in when
those operations are triggered. Memory-R1 updates memory after each turn, which is
a natural and reasonable design, while this work lets the model decide the timing in
an end-to-end way.
○ The point is that whether this flexibility is actually an advantage is not obvious.
I would like to see stronger empirical evidence, for
example, the proposed method triggers memory operations much less
frequently and therefore achieves higher efficiency, better utility, or broader
generalization across question types/domains to justify this design choice.
2. **Limited benchmark and model scope:** Evaluation is conducted only on a single
base model (Qwen3-Instruct) and two document-based QA datasets (NovelQA and
infiniteBench En.MC split). Broader long-term or multi-session benchmarks such as
LoCoMo, MSC, LongMemEval, or RULER (not necessarily all) should be included to
test richer memory behaviors.
3. **Lack of scalability and generalization tests:** The model is trained on the
*PublicDomain* split of NovelQA and mainly evaluated on the *Copyright* split. In
Table 3, its gains diminish on $\infty$Bench, particularly for larger backbones (e.g.,
Qwen3-8B). Additional experiments on cross-domain or OOD settings are needed to
assess the proposed method’s applicability, especially given its reliance on formatted
training.
4. **Missing baselines:**
○ Since the method equips the LLM with external memory, a fair comparison
should include the same base model with (i) direct function-call memory
access and (ii) MCP-based memory access (e.g.,
https://github.com/doobidoo/mcp-memory-service), to see whether the base
model (without SFT on reasoning traces) can already use memory when
given access, and how much extra benefit the proposed framework actually
provides.
○ Comparisons with prior memory-management methods, whether retrieval-
augmented (RAG) or agent-based methods such as those evaluated in Mem0
and Memory-R1, are necessary; For cost reasons, even a direct evaluation
within their setups would make the results more informative.

**Questions:**

5. What are the specific requirements for the “high-quality, good-behavior, expert
reasoning trajectories”?
6. Since the generation of expert reasoning trajectories is effectively a distillation
process from Claude-Sonnet-4, it would be informative to include Claude-Sonnet-4’s
own performance as a baseline in the Experimental section.

---

> ### Author Response · Authors · 2025-11-21
> **Response to Reviewer qeFC (1)**
>
> We thank the reviewer for highlighting the importance of stateful language modeling and for finding our “model as the wizard” paradigm “conceptually engaging”. Below we respond to each concern in detail.
>
> > **[W1] Comparison with Concurrent Work (Memory-R1)**
> >
>
> We appreciate the reviewer’s attention to our **concurrent** work Memory-R1 (released on **August 27, 2025**). We view the it as a partial validation of the importance of this research direction. We believe the Pensieve paradigm represents an at least equally significant contribution as Memory-R1, tackling the challenge through a unique and critical mechanism.
>
> **The Core Difference: Storage Update vs. Context Pruning**
> While both approaches involve "deletion", they operate on different levels:
>
> 1. **Memory-R1's deletion** operates on the *external memory bank*. It updates stored memories (which is functionally equivalent to our `updateNote` tool).
> 2. **StateLM's deletion** operates on the **actual context window**. It autonomously removes history tokens to manage cognitive load.
>
> Lastly, the table below shows that StateLM invokes memory operations (note, update, merge, read) **more selectively** (i.e., not at every round), which reduces unnecessary tool calls and lowers overall inference cost.
>
> | NovelQA | Memory Operations | Total Turns (Memory-R1-like Memory Ops.) |
> | --- | --- | --- |
> | StateLM-4B | 4.31 | 19.10 |
> | StateLM-8B | 4.46 | 19.35 |
>
> > **[W2, W3, W4] Model Scope, Benchmark Diversity, and Additional Baselines**
> >
>
> We agree that broader evaluation is important and have expanded both our models and our experimental scope. During this period, we have updated the search engine in our framework by improving the its frontend (tool results) with search previews and highlights, and upgrading the backend of using the more advanced ElasticSearch engine. We then retrained the models to obtain updated StateLMs. To strengthen the evaluation and address the reviewer’s concerns, we have incorporated the following points:
>
> - **Model Scope:** We add the **Qwen3-14B** model, which further improves the model scale. Note that we choose Qwen3 models mainly for their agentic capabilities, which lies the foundation for the StateLMs.
> - **Baselines**: We follow the reviewer’s advice to include prior memory-related methods and compare their performance to our StateLM, which includes
>     - *Mem-Agent [1]*: RL-trained LLM agents that process the long context in chunks and updates its memory context after reading each chunk.
>     - *ReadAgent [2]*: Workflow agent system that let the LLM decide what content to store together in a memory episode, compress those memory episodes into short episodic memories and take actions to look up passages in the original text complete a task.
>
>     |  | **NovelQA** | **InfiniteBench** |
>     | --- | --- | --- |
>     | RL-MemoryAgent-7B | 30.38 | 34.93 |
>     | RL-MemoryAgent-14B | 39.50 | 45.85 |
>     | ReadAgent-8B | 16.38 | 24.02 |
>     | ReadAgent-14B | 23.12 | 34.06 |
>     | StateLM-4B (updated) | 79.57 ± 0.53 | 69.58 ± 2.20 |
>     | StateLM-8B (updated) | 83.71 ± 1.28 | 69.58 ± 1.01 |
>     | StateLM-14B (new) | 85.47 ± 0.82 | 77.29 ± 0.44 |
> - **Evaluation Benchmark:** We follow the reviewer’s advice to include *LongMemEval* [3] in our evaluation, which benchmarks model’s long-term interactive memory capability. Due to the time constraint, we downsample the LongMemEval-S set by the question type to preserve the question difficulty and derive a subset of 50 problems to test our StateLM and the Qwen3-128K baseline on it.
>
>
>     |  | **LongMemEval-S (50)** |
>     | --- | --- |
>     | Qwen3-4B-128K | 46.67 ± 4.62 |
>     | **StateLM-4B-32K** | **58.67 ± 2.31** |
>     | Qwen3-8B-128K | 48.00 ± 4.00 |
>     | **StateLM-8B-32K** | **55.33 ± 4.16** |
>     | Qwen3-14B-128K | 60.67 ± 3.06 |
>     | **StateLM-14B-32K** | **70.00 ± 4.00** |
>
> The results show that enhanced search tool further improves the performance of larger model backbones such as Qwen3-8B and Qwen3-14B, and StateLM outperforms the evaluated baselines on more diverse datasets. Regrading the extra benefit from our framework, we provide a Qwen3 model with function call + system prompt approach and compare to our StateLM in Section 4.4. Together, these additional results provide a more complete picture of the robustness and breadth of the StateLM framework.
>
> [1] Yu, Hongli, et al. "MemAgent: Reshaping Long-Context LLM with Multi-Conv RL-based Memory Agent." *arXiv preprint arXiv:2507.02259* (2025).
>
> [2] Lee, Kuang-Huei, et al. "A Human-Inspired Reading Agent with Gist Memory of Very Long Contexts." *Forty-first International Conference on Machine Learning*.
>
> [3] Wu, Di, et al. "LongMemEval: Benchmarking Chat Assistants on Long-Term Interactive Memory." *The Thirteenth International Conference on Learning Representations*.

---

> ### Author Response · Authors · 2025-11-21
> **Response to Reviewer qeFC (2)**
>
> > **[Q1] Requirements for high quality expert trajectories.**
> >
>
> In our experiments, we apply several techniques to obtain high quality, good behavior expert trajectories. The key requirements are summarized below:
>
> - **Principled operations:** We design a detailed system prompt (shown in Figure 8) that instructs the teacher model to reason and act according to explicit process guidelines. This helps ensure that the generated trajectories follow a structured and interpretable workflow.
> - **Correct answer (Outcome):** We first apply outcome level reject sampling. Only trajectories with correct final answers are retained, and any trajectories containing API related errors or incomplete generations are removed.
> - **Timely deletion (Behavior):** We then examine whether the model removes note construction messages and readChunk outputs within the permitted number of rounds in the trajectory. Trajectories that fail to perform these deletions consistently are discarded, as such patterns may propagate to the student model and lead to inefficient context use or content length violations.
>
> Collectively, these requirements ensure that the retained training samples faithfully represent the behaviors needed for effective state management.
>
> > **[Q2] Claude-Sonnet-4 Performance Results**
> >
>
> We agree this is informative and provide Claude-Sonnet-4’s performance, along with an additional evaluation of Qwen3-235B, equipped with our Pensieve framework in the table below.
>
> |  | NovelQA | InfiniteBench |
> | --- | --- | --- |
> | Claude-4-Sonnet | 85.60 | 85.59 |
> | Qwen3-235B-A22B-Instruct-2507 | 80.85 | 72.37 |
>
> **You can also find the complete table of results in our general response.**
>
> ---
>
> We sincerely appreciate the time you took to provide such a detailed and constructive review. Your feedback was instrumental in motivating us to significantly strengthen the paper, particularly regarding the experimental scope and baselines.
>
> We understand that your initial lower rating (4) may stem primarily from concerns regarding the **novelty compared to Memory-R1** and the **breadth of evaluation**. In this response, we have systematically addressed these points:
>
> 1. **Clarified Concurrency & Distinct Mechanism:** We have clarified that Memory-R1 is a **concurrent work**. Furthermore, we demonstrated that the mechanisms differ substantially: StateLM features **Active Context Pruning** to handle the working load, whereas Memory-R1 focuses on updating an external memory bank.
> 2. **Expanded Empirical Evidence:** We have added **new baselines** (Mem-Agent, ReadAgent), **broader benchmarks** (LongMemEval), and **larger model scales** (Qwen3-14B), all of which confirm the robustness of our approach.
>
> Since we have directly resolved the specific weaknesses regarding novelty and evaluation scope that grounded your initial assessment, we respectfully request that you reconsider your rating. We are confident that the revised manuscript will present a comprehensive and significant contribution to the community.

---

### Official Review · Reviewer_vbKg · 2025-11-01

**Soundness:** 3
**Presentation:** 4
**Contribution:** 4
**Rating:** 8
**Confidence:** 3

**Summary:**

Stateful Language Models (StateLM or SLM) is a class of foundation models that are equipped with tools (including: dynamic indexing, context pruning, note taking) to manipulate their state in a reasoning loop which dynamically (and automatically) updates their context. A StateLM (i) can retrieve a needle in 1 million-token haystack (ii) in empirical results over practical QA benchmarks it performs better than strong instruct baselines using a fourth of their active context and (iii) is superior in learning to manage memory than agent-like prompting.

SLM's reasoning trajectory consists of a series of actions (thoughts and acts (tool invocations)) and states (optionally modified contexts and responses from tools): tools can do tasks like analyze text, summarize or search through it or even update (e.g. delete) context. SLM is trained over trajectories for handling questions of types involving either locating or understanding text (search or scan types): each trajectory consists of steps (training samples) where given the history up to some step, SLM is trained to predict next step's thought and action. Interestingly, SLM's performance cannot be matched by models that have access to tools and are prompted to follow the context management process in SLM.

**Strengths:**

- This is a simple and novel idea: the model becomes active, inspects its current memory/state and accordingly constructs the context to operate on using pre-defined tools.

- No pressing need for the user-in-the-middle role of building prompts conditioned on a manually inspected state (automation).

- Clean guidelines for training, set of orthogonal tools well-defined.

- Performance on long-context recall and QA benchmarks are impressive.

**Weaknesses:**

- Critical requirement for the availability of a strong LLM for the generation of training samples (in particular for process-mode classification)
- The set of tools is given, is generic enough but it certainly cannot fit any question handling.

**Questions:**

- Are there any thoughts for automatically building the set of tools most amenable to particular question types?

- Can the succession of the particular tools invoked drive the classification of questions answered into finer-grained classes?

---

> ### Author Response · Authors · 2025-11-21
> **Response to Reviewer vbKg (1)**
>
> We sincerely thank the reviewer for the positive assessment of our work, especially for finding the Pensieve idea “simple and novel” and the empirical results of StateLM “impressive.” We deeply appreciate the thoughtful questions and constructive feedback. Below we address the reviewer’s comments in detail.
>
> > **[W1] On the requirement for a strong LLM to create training trajectories**
> >
>
> We agree that a teacher model is needed to produce well-structured reasoning trajectories. Given the two main training paradigms of SFT and RL, our choice to adopt a teacher-guided SFT pipeline was a deliberate decision aimed at maximizing transferability and accessibility. This approach also aligns with widely adopted practices for training advanced reasoning models [1] and tool-use agents [2]. We highlight two advantages of this design:
>
> 1. **The Teacher is Flexible**: The "strength" of data generation stems less from the raw parameter scale and more from the sophisticated prompting coupled with our toolkit. This makes the pipeline accessible without strict reliance on SOTA closed-source models.
>
> 2. **Seamless Integration into Post-Training**:  the primary goal of this project is to evolve our foundation model itself, so we prioritized compatibility with standard post-training recipes. Our trajectories can be seamlessly mixed into general instruction-tuning datasets (e.g., alongside math or code) to endow foundation models with state-management capabilities. In contrast, RL requires complex reward engineering, making it difficult to integrate into the unified training pipelines of general-purpose LLMs.
>
> We view our current method as establishing a solid foundation for this new paradigm, demonstrating that the mechanism works effectively through the most reliable training path. Building on this foundation, we are excited to explore alternative learning approaches in the future work.
>
> [1] Guo, Daya, et al. "Deepseek-r1: Incentivizing reasoning capability in llms via reinforcement learning." *arXiv preprint arXiv:2501.12948* (2025).
>
> [2] Qin, Yujia, et al. "ToolLLM: Facilitating Large Language Models to Master 16000+ Real-world APIs." *The Twelfth International Conference on Learning Representations*.
>
>
>
> > **[W2] On the fixed toolset and extensibility & [Q1] Automatically constructing toolsets for different question types**
> >
>
> This is a brilliant suggestion that perfectly anticipates the future trajectory of our **Pensieve Paradigm**. We fully align with your perspective: **no static set of tools can serve as a "silver bullet" for every conceivable question.**
>
> To use an analogy fitting our paper's theme:
>
> - **From Textbook Spells to Spell Invention:**
> In the current iteration of the Pensieve Paradigm, our model acts as a **junior wizard**. It proficiently wields a set of "textbook spells" (our pre-defined tools) to manipulate memory. While effective, it is indeed limited to the "curriculum" we provided.
> We agree that the next step towards AGI, or "true wizardry", is enabling the model to **invent its own spells** (dynamically construct tools) when facing novel challenges.
>
> - **Technical Path to "Spell Invention":**
> While this is orthogonal to the current work, we are actively exploring specific directions to realize this vision:
>
>   - **Evolutionary Tool Synthesis:** Allowing the model to propose or refine tool definitions (e.g., via Python function generation) during training, effectively "meta-learning" custom spells for specific problem clusters.
>   - **Data-Driven Tool Discovery:** Exposing the model to a vast pool of potential actions and using clustering to identify fine-grained, task-specific subsets.
>
> We view this paper as establishing the **fundamental mechanics**. Once this foundation is solid, giving the model the autonomy to craft its own tools is the logical and necessary next step.

---

> ### Author Response · Authors · 2025-11-21
> **Response to Reviewer vbKg (2)**
>
> > **[Q2] Using the succession of tools to classify question types**
> >
>
> We appreciate this insightful suggestion. In fact, our trajectory data already indicates that tool sequences strongly correlate with the underlying task type:
>
> - **Search-type questions:** These questions can be answered by locating specific facts or entities in the document, typically triggered by concrete keywords or phrases. As a result, they lead to iterative use of `searchEngine`.
> - **Scan-type questions:** These questions require a broader or more sequential understanding of the text, such as summaries or reasoning that spans multiple locations. They naturally lead to repeated `readChunk` operations in sequence.
>
> During evaluation, StateLM also explicitly determines the question type and the corresponding process mode in the first round before proceeding. Given the minimalist nature of our current toolset, the classification is largely driven by `searchEngine` usage. We plan to incorporate richer taxonomies as we expand the toolset in future work.
>
> ---
>
> We thoroughly enjoyed reading your review. It is rare and encouraging to find a reviewer who so accurately grasps not just the technical details, but the **broader vision of the Pensieve Paradigm and StateLM**.
>
> We remain fully available to clarify any further details during the discussion period and have attached **expanded results in our general response**. If our responses have satisfactorily addressed your concerns regarding the training pipeline and toolset generalizability, **we would be honored to have your support during the discussion phase.** We believe this paper will serve as a valuable stepping stone for the community, and your backing would be instrumental in bringing it to the community.

---

### Author Response · Authors · 2025-11-21
**General Comments**

We sincerely thank all reviewers (vbKg, qeFC, GVg6, xaus) for their thoughtful feedback and for recognizing the importance of the problem, the "conceptually engaging" paradigm, and the "impressive" empirical results.
In response to the common concerns raised regarding **novelty, training methodology, toolset flexibility, and evaluation scope**, we have conducted extensive additional experiments and clarifications. We summarize our key responses below.

> **Novelty**: Reviewers (qeFC, GVg6) asked for clarification on how StateLM differs from prior agentic frameworks (e.g., Memory-R1).
>

We would like to revisit the Pensieve analogy to emphasize our core contribution. When Dumbledore’s mind becomes overburdened, he extracts memories into a Pensieve to revisit later. This act of “extraction” is the fundamental operation that allows the model to retain only the information needed for the next step, aligning directly with the objective of context engineering.

To the best of our knowledge, no prior work (including standard MCP agents) implements an operation comparable to `deleteContext`. The closest example is our concurrent study, Memory-R1 [1], whose deletion mechanism acts on an external memory bank rather than the working context itself. Thus, while our method may appear to be straightforward policy learning over a fixed toolset, we believe this is a case of **any mundane tool becoming a master's weapon**. The key outcome is that this design enables the model to learn to **manipulate its own state**.

[1] Yan, Sikuan, et al. "Memory-r1: Enhancing large language model agents to manage and utilize memories via reinforcement learning." *arXiv preprint arXiv:2508.19828* (2025).

> **Expert Trajectories or RL:** Reviewers (vbKg, qeFC, xaus) queried our choice of Supervised Fine-Tuning (SFT) on distilled expert trajectories and the possibility of using Reinforcement Learning (RL).
>

We truly appreciate this insightful observation on the trade-off between the stability of supervised synthesis and the autonomy of RL. Our choice was a deliberate strategic decision aimed at maximizing transferability and training compatibility, which aligns with widely adopted paradigms for training advanced reasoning models [1] and tool-use agents [2]. We highlight two key advantages:

1. **The Teacher is Flexible**: The "strength" of data generation stems less from the raw parameter scale and more from the sophisticated prompting coupled with our toolkit. This makes the pipeline accessible without strict reliance on SOTA closed-source models.

2. **Seamless Integration into Post-Training**:  the primary goal of this project is to evolve our foundation model itself, so we prioritized compatibility with standard post-training recipes. Our trajectories can be seamlessly mixed into general instruction-tuning datasets (e.g., alongside math or code) to endow foundation models with state-management capabilities. In contrast, RL requires complex reward engineering, making it difficult to integrate into the unified training pipelines of general-purpose LLMs.

Nevertheless, we fully agree that RL is the natural next step to further extend this framework, and we are excited to explore it in future work.

[1] Guo, Daya, et al. "Deepseek-r1: Incentivizing reasoning capability in llms via reinforcement learning." *arXiv preprint arXiv:2501.12948* (2025).

[2] Qin, Yujia, et al. "ToolLLM: Facilitating Large Language Models to Master 16000+ Real-world APIs." *The Twelfth International Conference on Learning Representations*.

---

### Author Response · Authors · 2025-11-21
**General Comment 2**

> **Fixed Toolset**: Reviewers (vbKg, GVg6, xaus) noted the limitations of a pre-defined, fixed toolset.
>

We fully align with your assessment. You are absolutely correct that **no static set of tools can serve as a "silver bullet"** for every conceivable complex scenario.

**From Textbook Spells to Spell Invention:** To use an analogy fitting our paper's theme: In the current iteration of the *Pensieve Paradigm*, our model acts as a **junior wizard**. It is learning to proficiently wield a set of "textbook spells" (our pre-defined tools) to manipulate memory. While this "curriculum" is sufficient to validate the effectiveness of the paradigm on current benchmarks, we acknowledge that dealing with highly dynamic real-world scenarios will eventually require **"true wizardry",** allowing the model to **invent its own spells** (dynamically construct tools).

We view this paper as establishing the **fundamental mechanics:** with a fixed toolset, the model can already learn to **dynamically** manipulate its own state. Once this foundation is solid, giving the model the autonomy to craft its own tools is the logical and necessary next step.

> **Expanded Evaluation:** To address concerns regarding fairness and scope (qeFC, GVg6, xaus), we have significantly expanded our experiments:
>

During this period, we first updated the search engine in our framework by improving the frontend (more informative search previews and highlights) and upgrading the backend to a stronger ElasticSearch pipeline. We then retrained our models and obtained updated StateLMs, including an **additional** **Qwen3-14B** variant.

We have also added several **new baselines**, including: (1) **Mem-Agent** [1]: an RL-trained agent that processes long contexts in chunks and updates its memory after each chunk; (2) **ReadAgent** [2]: a workflow-based system that learns to scan, compress, and retrieve episodic memories during long-document reasoning; (3) **Claude-4-Sonnet (200K)** equipped with our Pensieve framework, originally used for trajectory generation; and (4) **Qwen3-235B-A22B-Instruct-2507 (256K)**, a strong long-context baseline also integrated with our framework.

In addition, we incorporated a new benchmark, **LongMemEval-S** [3], which evaluates **interactive long-term memory capabilities**. Due to time constraints, we downsampled the dataset by question type to preserve difficulty and constructed a 50-problem subset to compare StateLM with the Qwen3-128K baseline.

The results shown in the table below are encouraging. The expanded evaluation scope provides strong evidence for the effectiveness of our method and addresses the concerns regarding fairness and breadth. We will update the complete experiments to our paper.

|  | **NovelQA** | **InfiniteBench** | **LongMemEval-S (new)** |
| --- | --- | --- | --- |
| Claude-4-Sonnet (new) | 85.60 | 85.59 | - |
| Qwen3-235B-A22B-Instruct-2507 (new) | 80.85 | 72.37 | - |
| RL-MemoryAgent-7B (new) | 30.38 | 34.93 | - |
| RL-MemoryAgent-14B (new) | 39.50 | 45.85 | - |
| ReadAgent-8B (new) | 16.38 | 24.02 | - |
| ReadAgent-14B (new) | 23.12 | 34.06 | - |
| Qwen3-4B-128K | 65.17 ± 0.53 | 59.97 ± 0.50 | 46.67 ± 4.62 |
| Qwen3-8B-128K | 65.87 ± 1.42 | 66.81 ± 1.16 | 48.00 ± 4.00 |
| Qwen3-14B-128K (new) | 77.94 ± 0.26 | 74.96 ± 0.25 | 60.67 ± 3.06 |
| **StateLM-4B-32K (updated)** | **79.57 ± 0.53** | **69.58 ± 2.20** | **58.67 ± 2.31** |
| **StateLM-8B-32K (updated)** | **83.71 ± 1.28** | **69.58 ± 1.01** | **55.33 ± 4.16** |
| **StateLM-14B-32K (new)** | **85.47 ± 0.82** | **77.29 ± 0.44** | **70.00 ± 4.00** |

[1] Yu, Hongli, et al. "MemAgent: Reshaping Long-Context LLM with Multi-Conv RL-based Memory Agent." *arXiv preprint arXiv:2507.02259* (2025).

[2] Lee, Kuang-Huei, et al. "A Human-Inspired Reading Agent with Gist Memory of Very Long Contexts." *Forty-first International Conference on Machine Learning*.

[3] Wu, Di, et al. "LongMemEval: Benchmarking Chat Assistants on Long-Term Interactive Memory." *The Thirteenth International Conference on Learning Representations*.

---

### Author Response · Authors · 2025-12-03
**Final Updates for Area Chair**

Dear Area Chair,

We sincerely appreciate your effort in handling this submission. To support efficient decision-making, we want to highlight **two critical issues addressed** in our rebuttal, in addition to our general comments and reviewer-specific rebuttals.

> **Comparison with Concurrent Work and Expanded Experimental Results**
>

**Reviewer qeFC (original rating 4)** questioned our design and requested comparison with **a concurrent work**, Memory-R1 (first released on August 27, 2025). In response, we clarify the timing of this concurrent work and explain our innovation of `deleteContext` for **active context pruning**, in contrast to the external memory updates used in Memory-R1. We further provide empirical evidence demonstrating our efficiency for autonomous memory operation over such fixed workflow. The reviewer also raised concerns regarding the experimental scope. In response, we enlarge the **model scope** (+ Qwen3-14B), expand **benchmark diversity** (+ LongMemEval), and include **additional baselines** (+ ReadAgent, MemAgent, Claude-4-Sonnet, and Qwen3-235B-A22B-Instruct). The results further validate the effectiveness and robustness of our approach.

We believe we have fully addressed the reviewer’s concerns, and reviewer qeFC would be delighted to see these additional clarifications and results.

> **Expert-trajectory SFT is a Deliberate Choice**
>

Several reviewers queried our choice of Supervised Fine-Tuning (SFT) on expert trajectories and the possibility of using Reinforcement Learning (RL). Our choice is a deliberate decision aimed at maximizing transferability and training compatibility. In practice, the teacher model is **highly flexible** due to our principled system prompt and toolkit, and the resulting training samples can be **seamlessly integrated into the post-training** pipeline alongside general instruction-tuning and reasoning data. In contrast, RL requires complex, task-specific reward engineering, making it difficult to integrate into a unified training pipeline for general-purpose LLMs.

As noted by the reviewers, the Pensieve paradigm and the resulting StateLMs address a crucial problem in building real-world agentic models, enabling effective long context handling and autonomous context (memory) management. We believe this provides a novel and effective path towards future agentic foundation models. Thank you for your time and fair evaluation during this special period.

Best regards,

Authors

---

### Meta-Review · Area_Chair_QTqX · 2026-01-06

**Summary:**

This paper introduces StateLM, a stateful language model framework inspired by the "Pensieve" analogy, enabling models to actively manage memory via learned tool use (indexing, pruning, note-taking). The authors demonstrate strong performance on synthetic and real-world long-context QA tasks, outperforming fixed-window baselines with significantly reduced context usage.

Reviewers appreciated the clear presentation, strong empirical results, and the important problem addressed. However, several key concerns were raised regarding novelty, evaluation scope, and implementation limitations. The rebuttal has addressed most of the concerns. Overall, the strength of the paper overwhelms the weaknesses and the paper is recommended for acceptance.

PS: The following reference does not exist and authors are required to cite the correct source in the revision.
Andrej Karpathy. Context engineering is the delicate art and science of filling the context window... https://x.com/karpathy/status/1785482721150992423, Apr 2025. Accessed: 2025-06-25.

**Reviewer Concerns:**

Key Concerns Raised:

- Novelty vs. prior work (qeFC, GVg6): How does StateLM's "deleteContext" differ from prior memory eviction mechanisms (e.g., MemGPT, Memory-R1)?

- Limited evaluation scope (qeFC, GVg6): Benchmarks focused primarily on QA; missing diverse baselines and broader task evaluation.

- Fixed toolset (vbKg, GVg6, xaus): Predefined tools may not generalize to complex scenarios.

- Training methodology (xaus, vbKg): Why SFT instead of RL? Dependence on strong teacher LLM for trajectories.

- Baseline fairness (xaus): Comparison may not be apples-to-apples; suggested sliding-window alternatives.

Addressed in Rebuttal:

- Clarified mechanistic distinction from Memory-R1 (context pruning vs. memory bank updates)

- Added multiple new baselines (MemAgent, ReadAgent, Claude-Sonnet)

- Extended model scale (Qwen3-14B) and benchmark diversity (LongMemBench)

- Provided rationale for SFT choice (compatibility, accessibility)

- Implemented MemAgent as sliding-window baseline

Outstanding/Partially Addressed:

- Novelty debate persists

- Toolset remains fixed (acknowledged as future work)

- Generalization beyond Qwen family not demonstrated

**Reviewer Scores:**

vbKg: 8 → Likely 8

qeFC: 4 → Likely 6 or 4

GVg6: 4 → Likely 6 or 4

xaus: 6 → Likely 6 or 8

---

### Decision · Program_Chairs · 2026-01-26

Accept (Poster)